# Factors Associated with Smoke-Free Pregnancy among Aboriginal and Torres Strait Women and Their Experience of Quitting Smoking in Pregnancy: A Mixed Method Cross-Sectional Study

**DOI:** 10.3390/ijerph182111240

**Published:** 2021-10-26

**Authors:** Tabassum Rahman, Amanda L. Baker, Gillian S. Gould, Kerrin Palazzi, David Lambkin, Michelle Kennedy

**Affiliations:** 1School of Medicine and Public Health, The University of Newcastle, Newcastle, NSW 2308, Australia; amanda.baker@newcastle.edu.au (A.L.B.); michelle.bovill@newcastle.edu.au (M.K.); 2Hunter Medical Research Institute, Newcastle, NSW 2305, Australia; kerrin.palazzi@hmri.org.au (K.P.); david.lambkin@hmri.org.au (D.L.); 3Faculty of Health, Southern Cross University, Coffs Harbour, NSW 2450, Australia; gillian.gould@scu.edu.au

**Keywords:** Aboriginal and Torres Strait Islander women, smoking cessation, smoke-free pregnancy, Aboriginal Health Services, indigenous-led health research

## Abstract

Smoke-free pregnancies have long-term health benefits for mothers and babies. This paper quantitatively examines factors associated with smoke-free pregnancies among Aboriginal and Torres Strait Islander women (hereafter Aboriginal women) and qualitatively explores their smoking cessation (SC) experiences during pregnancy. An Aboriginal-led online cross-sectional study on SC was conducted with Aboriginal women and in partnership with Aboriginal communities, between July and October 2020. The present analysis includes participants who made a pregnancy-related quit attempt (N = 103). Chi-squared tests, logistic regression models, and thematic analysis of free-form text responses were performed. The adjusted odds of having smoke-free pregnancies were 4.54 times higher among participants who used Aboriginal Health Services (AHS) (AOR = 4.54, *p*-value 0.018). Participants living in urban settings had 67% lower odds of having smoke-free pregnancies compared to their regional/remote counterparts (AOR = 0.33, *p*-value 0.020). Qualitative data revealed strong motivations to reduce tobacco-related harms to the fetus and variability in quitting experiences at different stages of and across pregnancies. Smoking cessation care (SCC) can support Aboriginal women meaningfully if their quitting experiences are considered in SCC development and implementation. Consistent funding for AHS-led SCC is needed to garner health benefits for Aboriginal peoples. More research into urban versus regional/remote differences in maternal SC is recommended.

## 1. Introduction

Smoking in pregnancy remains a major public health concern. First Nation peoples in high-income countries, namely, the United States of America (USA), Canada, New Zealand, and Australia, are disproportionally affected by tobacco smoking [1], with current health inequity being related to colonisation. Smoking during pregnancy is more common among First Nation women compared to women in the general population in high-income countries [1]. Social disadvantage, stressful life events, lack of access to culturally appropriate smoking cessation support, and a high prevalence of smoking in First Nation communities, owing to the legacy of colonisation instilled into public policies, are some of the major contributing factors to persistent smoking during pregnancy [1,2,3]. First Nation women are thus at higher risk of smoking-related adverse pregnancy outcomes, including placenta previa, preterm birth, intrauterine growth restriction, spontaneous abortion, and stillbirth [4]. It also elevates the risk of low birth weight, respiratory problems and related hospitalisation, behavioural problems in children, and risk of obesity and hypertension in early adulthood [4,5]. However, quitting smoking because of pregnancy was found to be common among First Nation women [1].

In Australia, Aboriginal women are highly motivated to quit smoking, with many attempting to quit during pregnancy to give their baby the best start in life [6]. Although 42% of Aboriginal women aged 18 or more smoke tobacco in any amount [7], over the past decade, the proportion of Aboriginal women who smoked during pregnancy declined by approximately eight percentage points (44% in 2017 vs. 52% in 2009) [8]. Nevertheless, existing evidence demonstrates barriers that Aboriginal women face in quitting [9,10,11,12] smoking, for instance, difficult life circumstances, social cues, and lack of support. On the contrary, high motivation to give up smoking and resilience and support from family and community are major enablers to smoking cessation among Aboriginal women [2]. Aboriginal women make multiple quit attempts during pregnancy [13]. However, compared to the general population, fewer Aboriginal women are able to sustain abstinence [9,14]. While the barriers faced by women in the general population are similar, Aboriginal women are likely to be disproportionately affected due to the ongoing impacts of colonisation, dispossession, and racism [2]. Nevertheless, qualitative evidence suggests that Aboriginal women may be able to achieve long-term abstinence via repeated quit attempts [11].

Giving up smoking during pregnancy has significant long-term health benefits for mother and child [15]. Quitting even for a short time is beneficial to the growth of the fetus [16]. However, there is a paucity of evidence highlighting Aboriginal peoples’ progress in health. Reporting of evidence on tobacco smoking in these contexts is often devoid of consideration of health inequity based on race and ethnicity and tends to take a deficit approach to First Nation peoples’ health [17].Leaving the health progress unexplored and unrecognised has direct implications on how Aboriginal peoples’ health needs are perceived, policies are shaped, and resources allocated. When evidence generated from a deficit point of view feeds into health interventions, it is likely to influence improvements in Aboriginal peoples’ health outcomes.

The existing literature presents evidence on Aboriginal and Torres Strait Islander women’s (hereafter ‘Aboriginal women’, with acknowledgement of autonomy of all Aboriginal and Torres Strait Islander peoples) experience of smoking cessation with a focus on their knowledge and perception regarding harms of smoking during pregnancy [9,10,11,12,14,18,19]. However, they offer little insight into what helps Aboriginal women to quit smoking during pregnancy and their quitting behaviour. This study was conducted by an Aboriginal-led research team in partnership with Aboriginal communities in urban and regional New South Wales (NSW), Australia. It seeks to examine the progress that has been taking place in Aboriginal women’s health by examining factors associated with smoke-free pregnancy. Leveraging the factors that enable smoke-free pregnancies will provide guidance for future policies and resource allocation in improving one of the key health priorities in Australia and thus, will contribute towards increased life expectancy of Aboriginal peoples. The evidence presented in this paper may also be relevant in addressing smoking during pregnancy among First Nation women in other high-income countries. To address this gap in the evidence, this paper aims to (a) quantitatively examine the factors associated with smoke-free pregnancies among Aboriginal women in Australia and (b) qualitatively describe women’s self-reported experiences of quitting during pregnancy.

## 2. Materials and Methods

This paper reports on several pregnancy-related questions from an online survey of smoking and quitting among Aboriginal women. The parent project, entitled ‘Which Way?’, aimed to co-develop an Aboriginal-led evidence base for smoking cessation interventions to support Aboriginal women to be smoke-free during pregnancy and beyond and has been reported elsewhere [20]. The Which Way? Survey aimed to report the nature and characteristics of smoking and quitting behaviours of Aboriginal women aged 16 years and above. The study was led by an Aboriginal researcher and was developed in co-ownership with Aboriginal communities in urban and regional NSW.

The cross-sectional survey was conducted online due to the major impact of COVID-19 and the restrictions concerning face-to-face data collection. Data were collected between July and October 2020. The Research Electronic Data Capture (REDCap) application was used for the collection, management, and storage of data [21,22]. Participants were able to fill out the survey using a mobile phone, tablet/iPad, or a laptop/computer via an open-access link to the survey. Informed consent was digitally obtained at the beginning of the survey. Participation was only allowed upon providing consent.

The parent survey included a total of 78 items. The present paper reports data from 24 of the survey items pertaining to demographic characteristics, smoking-related variables, use of Aboriginal Health Services (AHS), and remoteness of location. AHS or Aboriginal Community Controlled Health Services are the main provider of Aboriginal primary health care in Australia. AHS are key players in ensuring improvement in Aboriginal health and wellbeing and developing the health of Aboriginal communities. Clinical service, health promotion, cultural safety, and community engagement are some of the major components of the distractive model of care of AHS [23]. Smoking and quitting history-related questions were specifically developed for the survey in collaboration with the Aboriginal Governance committee. Smoking questions covered the level of tobacco consumption, and the Heaviness of Smoking Index (HSI) was used to measure nicotine dependence [24]. The HSI is a reliable measure of nicotine dependence and has been utilised in studies conducted with Aboriginal peoples before [25,26]. Data were also collected on the frequency of urges to smoke (FUTS) (in the last 24 h) and strength of urges to smoke (SUTS) [27], which have previously been measured in Aboriginal communities [26]. Quitting history questions covered quit attempts in the context of pregnancy and otherwise. Questions included the time since the latest quit attempt, nature of the last quit attempt (sudden stop versus gradual reduction), duration of the longest quit attempt, and history of smoking cessation medication use. Qualitative data were gathered from the free-form text option where participants could provide more information on their quitting experiences in the pregnancy context. Remoteness of location was determined from participants’ postcode, according to the Australian Statistical Geographic Standard used by the Australian Bureau of Statistics (ABS) [28]. Appendix A provides further detail of the construction of variables analysed in this paper.

### 2.1. Sampling

The online sampling strategy used Facebook and Instagram platforms. Our primary points of recruitment included Aboriginal community partners who played a key role in promoting the study, and study team members promoted the study on their social media pages to drive recruitment. These primary points were used to facilitate the provision of project information, helping to recruit participants via convenience and snowball sampling. Eligibility was determined via screening questions regarding age, ethnicity, and smoking status asked at the beginning of the survey. Women who were younger than 16 years, non-Aboriginal or Torres Strait Islander, and/or who never smoked were ineligible. The sample size was informed by Ogundimu et al. (2016) [29] and calculated based on a minimum of ≥10 events per variable. Given that the variable of interest was quitting during pregnancy, a minimum sample size of 84 Aboriginal and Torres Strait women was required (i.e., 12% of expecting Aboriginal and Torres Strait women quit smoking during pregnancy [8]; therefore, to reach ≥10 events per variable, we required a minimum of 84 participants). The survey ended for potential participants who were deemed ineligible based on their responses to the screening questions (Figure 1).

### 2.2. Data Analysis

This analysis compares Which Way? participants who reported having at least one smoke-free pregnancy versus those who reported not having a smoke-free pregnancy. However, participants with multiple pregnancies who reported being both smoke-free in one pregnancy and continued to smoke in another were excluded from tests of associating and logistic regression models to aid the analysis and avoid misclassification of outcomes. To define a smoke-free pregnancy, participants were asked: “Were any of your quit attempts because you were pregnant?” Participants replying in positive were then asked: “Were you able to stay smoke-free the whole way through your pregnancy?” The response options were: (a) Stayed smoke-free whole way through; (b) Smoked occasionally; (c) Stayed smoke-free for several months; (d) Cut down; and (e) Others. Multiple responses were allowed to capture quitting history in different pregnancies. Participants who chose the response option (a) were considered to have a smoke-free pregnancy, while those who chose the response options between (b) and (e), for any pregnancy, were collapsed into one category as not having a smoke-free pregnancy. To determine factors associated with a smoke-free pregnancy, participants who chose response options (a) and any other options between (b) and (e) for different pregnancies were excluded. In the free-form text option, participants were requested to provide more information on their pregnancy-related quitting attempts. Participants were included in the analysis independent of the number of children living in their house or current pregnancy. They could type in as much information as they wanted to share regarding their quitting experiences. Those responses have been used as qualitative data in this analysis.

To determine geographical remoteness, participants’ postcode locations were initially grouped into five geographical categories: very remote; remote; outer regional; inner regional; and major cities, according to the Australian Statistical Geography Standard [28]. Based on the distribution of the participants across geographical locations, these categories were then rearranged into two broad categories to aid regression modelling (i.e., major cities versus inner regional, outer regional, remote, and very remote) and renamed urban and regional/remote settings. No cross-category postcode was found in this subsample. Stata V.15 statistical package was used for the quantitative analysis [30].

Descriptive characteristics of the sample associated with smoke-free pregnancy were examined. Pearson’s chi-squared test or Fisher’s exact test (if low cell numbers) were performed to compare response proportions between participants who reported having a smoke-free pregnancy and those who did not. Differences in means in continuous variables, i.e., age and number of cigarettes one smokes per day between these groups were measured by performing student’s *t*-tests and Mann–Whitney *U*-tests depending on the distribution of the data and number of observations. Student’s *t*-test was used to estimate the mean age of the participants, while Mann–Whitney *U*-test was used to estimate the mean number of cigarettes smoked daily by those who identified as smokers. The Alpha level was set at 5%. Associations between factors of interest and a smoke-free pregnancy were determined by performing univariate and multivariable logistic regression analyses. The equality of multiple coefficients was tested for factors with three or more categories, where statistical nonsignificance indicated no statistical difference in coefficients for different categories with the variables. Where the test result was nonsignificant, pairwise tests between variable levels were not performed to reduce type I error.

The following characteristics of interest were tested for association with smoke-free pregnancy: age; education; use of AHS; geographic remoteness; time since last quit attempt; longest quit attempt; nature of last quit attempt; history of trying smoking cessation medication; reduction in smoking in the month before the survey; the number of cigarettes smoked per day; HSI score; SUTS; and FUTS. Statistical and clinical significance was considered while including the variable in regression models. Although statistically nonsignificant in the chi-squared test, following guidance from relevant literature, age and education were included in the regression models [31], along with use of AHS [23] and remoteness [32]. Given the importance of smoking cessation medication in giving up smoking, the association between previous use of such medication and smoke-free pregnancies was explored.

Finally, a thematic analysis of qualitative data was performed. The qualitative data was coded independently by the first author of the paper (T.R.) who is a non-Aboriginal researcher and by the senior author of the paper (M.K.) who is an Aboriginal researcher. Any discrepancies in coding performed by TR and MK were resolved through discussion. TR performed a higher order thematic analysis with frequent reflexive discussion with M.K. NVivo software V.12 for analysing qualitative data was used to code and analyse the qualitative data [33]. Member counting for each theme was also performed.

## 3. Results

### 3.1. Participation Description

A total of 103 women reported attempting to give up smoking because of pregnancy (Figure 1). Of these, 46 (44.66%) reported having smoke-free pregnancies, while 57 (55.34%) participants smoked during any of their pregnancies.

Table 1 describes the characteristics of the subsample. Participants were predominantly Aboriginal participants (95.15%) and from NSW (45.63%), followed by Queensland (QLD) (28.16%) and Victoria (VIC) (11.56%). The mean age of participants was 32.13 years (sd ± 7.47). The majority of the participants (33.98%) completed up to Year 11 at school, followed by 27.18% of participants completing a Technical and Further Education (TAFE) certificate or university degree. Three quarters of the participants reported that they had access to an AHS. Just over half of the participants (56.31%) lived in urban settings while the rest of the participants were from regional/remote settings. About half of the participants had 1–2 children living in the household, and about 40% had three or more children living in the household. Six participants reported being pregnant at the time of the survey.

### 3.2. Factors Associated with Smoke-Free Pregnancy

Of the 46 participants reporting smoke-free pregnancies, three reported that they continued to smoke during at least one of their pregnancies. Thus, they were excluded from the analyses performed to determine factors associated with smoke-free pregnancies. Table 2 compares the characteristics for participants reporting having smoke-free pregnancies versus not.

Statistically significant differences were seen between those who had smoke-free pregnancies versus those who smoked cigarettes in terms of AHS use and remoteness. Univariate and multivariable logistic regression analyses were performed to further explore the association between these and other factors and having a smoke-free pregnancy (Table 3).

After performing multivariable logistic regression on age, education, previous use of smoking cessation medications, and use of AHS and geographical remoteness, AHS use (AOR = 4.54, 95% CI of 1.29–15.95, *p*-value 0.018) and geographical remoteness (AOR = 0.33, 95% CI of 0.33–0.84, *p*-value 0.020) remained independently associated with a smoke-free pregnancy. The analysis was adjusted for age, education, and previous use of smoking cessation medications. Participants living in urban settings had 67% lower odds of maintaining abstinence in a pregnancy-related quit attempt when compared to participants living in regional/remote settings. Although not statistically significant, participants in the older age groups appeared to have a higher chance of staying smoke-free during pregnancies compared to participants aged between 16–25 years. Similar trends were identified regarding higher education and having tried smoking cessation medication. The test of equality of multiple coefficients was performed for age (*p*-value = 0.703) and education (*p*-value = 0.588), as these variables have three or more categories. The results indicated that responses within these variables were statistically similar, and pair-wise comparisons were not further interpreted.

### 3.3. Women’s Experiences of Smoking Cessation during Pregnnacy

The qualitative analysis included the total subsample (N = 103) whether women reported a smoke-free pregnancy or not. The data pertain to experiences of changes in smoking behaviour before or in preparation for, during, and/or after a pregnancy. The qualitative data revealed that pregnancies influence the smoking behaviour of Aboriginal women. The themes and subthemes are summarised in Table 4. Appendix A provides more detail on participants quitting experiences during pregnancies.

#### 3.3.1. Motivations for Quitting Smoking

*Making quit attempts after being aware of the pregnancy*: Many Aboriginal women reported stopping smoking as they were motivated to quit smoking to protect their unborn baby from tobacco-related harms and to give them the best start in life. Some women stopped smoking as soon as they came to know about their pregnancy.
“As soon as I heard the baby’s heart beat I quit.”37 year old, stayed smoke-free

*Morning sickness made smoking less feasible*: Many of the participants could not stand the smell of a cigarette or would want to vomit if they attempted to smoke a cigarette due to morning sickness. While this is a physical condition that the women experienced, it enabled staying smoke-free during their pregnancy to a varying degree.
“My morning sickness was aggravated by the smell of smoke so I didn’t smoke and couldn’t be around smokers.”33 year old, smoked occasionally

#### 3.3.2. The Levels of Behavioural Change May Vary

*Reduction in consumption*: Many Aboriginal women mentioned that they reduced their consumption of cigarettes significantly during at least one of their pregnancies, an intention to minimise harms to their unborn child. Some of the participants mentioned that stopping smoking completely was difficult for them as smoking would help them cope with daily stress. Some, on the other hand, smoked occasionally due to stress.
“I was too stressed and decided it would be better for me to cut down instead.”42 year old, cut down

*Complete cessation*: Many of the participants who stopped smoking completely had not used any aids to quit, meaning they quit cold turkey. Some of the participants gradually reduced their tobacco consumption and quit.
“It was hard at first so I cut down in the beginning until I finally quit. It made it easier to quit knowing I was growing a bub.”19 year old, stayed smoke-free

Relatively long-term abstinence was also observed. Six participants maintained abstinence for a long time after delivery and started to smoke again when their child was a toddler.
“I quit as soon as I found out [I was pregnant] with my first and didn’t start again until my 3rd child was 3.”35 year old, stayed smoke-free

#### 3.3.3. The Experience of Change Is Diverse, either Difficult or Easy

*Giving up smoking can be difficult or stressful:* Many participants found giving up smoking during pregnancy to be difficult or stressful. Smoking was a way to cope with daily stressors for some of the participants. On some occasions, life circumstances, including being in an abusive relationship, made quitting smoking harder. However, some participants reported that they overcame the difficulty by thinking of the health of their babies. A few participants who found giving up smoking difficult or stressful mentioned feeling agitated or ‘stressing more’ after giving up, potentially indicating withdrawal symptoms. A few other participants mentioned that being around other smokers made quitting difficult.
“It was hard at first but stopped after 3–4 months into pregnancy and only smoked when I was stressing out.”17 year old, smoked occasionally and remained smoke-free for months

Not having cigarettes seemed to make coping with stress harder. Such an understanding received strength from unsubstantiated advice from health professionals (HP) that quitting smoking might have an adverse effect on the fetus via excessive stress.
“With my first I was advised to continue to smoke as trying to quit was putting too much stress on my body and could cause loss of baby, I was told to cut down, which I did. Second pregnancy I quit altogether and lost my twins at 18 weeks, 3rd and 4th I quit in the last few months of pregnancy.”33 year old, remained smoke-free for months and cut down

*Some Aboriginal women may find quitting smoking easy during pregnancy*: for a desire to protect their child from tobacco-related harms made quitting easy for some participants.
“Quitting while pregnant was easy, just wanted to give my child the best start to life.”23 year old, stayed smoke-free

#### 3.3.4. Experience of Change Evolves over Time and in Different Pregnancies

*Change in smoking behaviour over the course of a pregnancy*: Change in the smoking behaviour appears to not always be linear; change can happen in any direction. Some of the participants who gave up smoking at the beginning of a pregnancy smoked occasionally later in their pregnancy to ease off stress. On the other hand, some participants gradually reduced their daily consumption of cigarettes before they stopped smoking in the later part of their pregnancy.
“I quit for the first few months, I started stressing at work midway through my pregnancy and started smoking again.”22 year old, smoked occasionally

*Smoking and quitting experiences may vary in different pregnancies*: Change in smoking behaviour can be specific to individual pregnancies for some women. Participants shared how their experiences varied over multiple pregnancies, some reporting greater difficulty in quitting during subsequent pregnancies and during others, long-term abstinence lasting for two to three years after delivery.
“Its hard to do whilst pregnant. First and second I smoked most the way through cutting down. Third I gave up for months but not all. Last I gave up before getting pregnant.”35 year old, stayed smoke-free, remained smoke-free for months, and cut down

## 4. Discussion

The results from an online survey of 103 Aboriginal women offered important insights into Aboriginal women’s experiences of quitting smoking in the context of pregnancy and the factors that influence pregnancy-related quit attempts. Having smoke-free pregnancies was found to be significantly associated with using an AHS and living in regional/remote settings.

Qualitative data showed Aboriginal women in this study were motivated and wanted to have smoke-free pregnancies for the wellbeing of their babies. Women’s experience of smoking cessation varied over the duration of a pregnancy and across different pregnancies.

AHS play a critical role in providing primary and preventive health care to Aboriginal communities across Australia [23]. Over 80% of the clients who received care at AHS are Aboriginal peoples [34]. Cultural safety critically influences Aboriginal use of health care services [34]. AHS have been well regarded for their competence in creating a culturally safe and respectful space for Aboriginal people to receive SCC [35,36]. Cultural respectfulness, a relationship well established on trust, and an empowering approach were found to be important prerequisites for acceptance of SCC [37]. Chamberlain et al. (2017) highlighted the significance of cultural respect as one of the critical strategies in order for tobacco control initiatives to be meaningfully beneficial to Aboriginal health [38]. However, the present study did not capture any evidence of AHS use during pregnancy. Therefore, the association in our study may suggest an overall impact of the wider health promotion, messaging, and support that Aboriginal women received over their lifetime from AHS.

The smoking cessation rate during pregnancy is often measured based on the proportion of Aboriginal women smoking in the first 20 weeks of pregnancy, according to Australian national core maternity indicators [39]. Based on this indicator, 12% of the Aboriginal women who smoked gave up smoking during pregnancy [8]. Of the participants in the Which Way? Cohort, about 30.84% attempted to give up smoking at some stage during their pregnancy (Figure 1). While this study did not report quitting at specific time points, our findings exceed the national smoking cessation rate (22%) during pregnancy, in general [8]. Approximately 45% (Figure 1) of those who made a quit attempt because of a pregnancy remained entirely smoke-free during their pregnancies. Of the participants who did not stay smoke-free for the entire time of their pregnancy, about 55% changed their smoking behaviour to reduce tobacco-related harms to their unborn babies. This means that about 93% of Aboriginal women changed their smoking behaviour during pregnancy. This is the first time that changes in smoking behaviour have been reported, rather than quitting as a binary outcome. This is significant as it highlights the need for health professionals to support Aboriginal women’s willingness to manage their smoking during pregnancy with effective SCC [40]. The qualitative evidence offers important insight into Aboriginal women’s experiences of giving up smoking in pregnancy. This is the first study to offer qualitative evidence on Aboriginal women’s quitting behaviour across different pregnancies, drawn from over a hundred women, and across geographical settings.

The qualitative data reveal that Aboriginal women made quit attempts at different stages of their pregnancies. Only measuring quitting attempts that happen in the first 20 weeks of gestation thus does not account for women who may have quit later in the pregnancy and continued to stay abstinent, which overlooks the health benefits of quitting later in pregnancy [15]. This is particularly true when Aboriginal women make multiple quit attempts at different stages of pregnancy to give up smoking [13]. Previously, Gould et al. (2017) reported that Aboriginal women made repeated quit attempts in subsequent pregnancies and eventually achieving long-term cessation [11]. Conventional styles of reporting also do not take into account the positive changes that Aboriginal women make in their smoking behaviour, as demonstrated by the qualitative analysis in this study. Our study shows that women make multiple quit attempts to achieve cessation and adopt other strategies, such as smoking only occasionally, quitting smoking for several months, and reducing consumption of cigarettes to minimize tobacco-related harms to their unborn child. In line with Askew et al. (2019), a large number of women in this analysis found quitting to be stressful and difficult while grappling with difficult life circumstances [41]. Repeated attempts to quit and trying different strategies are indicative of agency, strength, and resilience [2]. Small-scale smoking cessation interventions that took into account Aboriginal women’s lived experiences regarding smoking in pregnancy were able to support women who were willing to give up smoking during pregnancy [41,42]. Thus, there is a need to modify the standard approach in reporting evidence on smoking cessation outcomes among Aboriginal women to one that is more equitable and inclusive in nature. There is also a need to take into account psychosocial circumstances, which are often related to colonisation.

Aboriginal women appreciate initiatives that recognise and respect their agency [43] and enable ownership of their own smoking cessation journey [37]. Engagement of Indigenous health workers appears to be a key element in effective SCC among First Nation peoples in high-income countries with histories of colonisation [36]. Systemic changes, such as enhancing Aboriginal leadership in research policy and practice, including increasing the number of Aboriginal SCC providers [2] and enhancing the capacity of the smoking cessation workforce [38], can potentially improve smoking cessation outcomes for pregnant Aboriginal women. Aboriginal women prefer a proactive but unimposing and nonjudgmental way of conversation with their health professionals around smoking cessation [2]. According to Bar-Zeev et al. (2017), there is scope to further improve provision of effective SCC among both Aboriginal and non-Aboriginal health professionals and increase their evidence-based knowledge, skills, and confidence [40]. The need for better training for health professionals is highlighted in the qualitative findings. For instance, some participants who reported high stress levels were advised by their doctors to continue to smoke and cut down because too much stress may cause loss of pregnancy. This gives strength to the evidence that there is a need and a desire for more training for health professionals [12,44].

Participants living in urban settings being less likely to have smoke-free pregnancies offers new insight into how living in regional/remote settings influence smoking cessation experiences of pregnant Aboriginal women. Other evidence shows that fewer women smoke during their pregnancies in major cities (38%) compared to remote (48%) and very remote areas (55%) [8]. Generally, in high-income countries, such as the USA and Australia, the prevalence of tobacco smoking is likely to be higher in non-urban areas than urban areas [32,45]. In Australia, access to health care services are not equally dispersed across the country, thus, people living in regional/remote areas are likely to have poorer health outcomes and less access to services [32]. Additionally, SCC modes and provisions may vary across different locations according to context [46]. Nevertheless, in this sample, participants living in regional/remote settings had more success in staying smoke-free through their pregnancies. This is a new finding that is in contrast with the current literature. However, this study was conducted online and thus, restricted to participants with internet access, which may have biased the findings. Therefore, this finding needs to be interpreted with caution and investigated further.

Although not statistically significant, women in older age groups appear to have higher odds of staying smoke-free through the entire pregnancy. This in line with existing evidence in the general population that smoking during pregnancy is commonly found among the younger cohort of women [4]. Likewise, participants with higher educational levels had statistically non-significant increased odds of staying smoke-free through to the end of their pregnancies. Less education was found to increase the likelihood of high nicotine dependence [47,48] and lower the chances of quitting [49]. Future research could explore the association between these factors and a smoke-free pregnancy. This highlights the scope for further exploration in future research.

### 4.1. Strength and Limitations of the Study

This mixed methods study was co-developed via thorough consultation with Aboriginal communities, including the development of questions asked to generate evidence and governance of the study. Whilst previous papers have focused on reporting smoking status at 20 week gestation, the present study takes a more inclusive approach, taking into account smoking cessation that happens after 20 weeks and other changes in women’s smoking behaviour that benefits the health of mothers and babies. Identification of characteristics that positively influence smoke-free pregnancies will contribute to the development of improved SCC for Aboriginal women and to positive pregnancy outcomes [4]. This will potentially contribute to improving the low uptake of SCC [50]. Thus, facilitating such positive change in women’s smoking behaviour is likely to have immense intergenerational health benefits for the future generations of Aboriginal peoples and increasing their life expectancy towards the highest achievable level. Leveraging the factors that enables smoke-free pregnancy will provide guidance for future policies and resource allocation in improving one of the key health priorities in Australia.

The study was conducted online due to COVID-19 related restrictions on face-to-face data collection. This may have limited the sample with regard to access to internet. However, most Aboriginal people are social media users, with 74% of Aboriginal people being connected to social media [51]. Nevertheless, this may have some implications for the representativeness of the sample [52]. This is not a population representative sample. However, this study offers important evidence on Aboriginal women of reproductive age group who represent less than 1% of the Australian population [53]. Previous nationally representative studies, such as Mayi Kuwayu, report a 2.3% response rate with the highest respondents being over 50 years of age [54]. Acknowledging the challenges in engaging Aboriginal women of reproductive age in research, this study provides novel findings from a sufficient sample relevant to the main outcomes. The survey may have elicited responses from participants who were more motivated to quit smoking, which may have influenced a higher quitting rate in this subsample. The qualitative responses vary in terms of the comprehensiveness of information they provide. Being an online survey, the sample was limited by connectivity. The evidence reported in this study is limited by the small sample size. Therefore, the association between remoteness of location and smoking-free pregnancy needs to be explored further with a larger sample. Moreover, participants from remote and very remote areas were not well represented in the survey, possibly due to lack of connectivity in those areas.

### 4.2. Implications for Policy and Practice

AHS are strategically best placed to support Aboriginal women in giving up smoking during pregnancy with a holistic approach and embedded connection to culture, land, and community. Therefore, sustained, long-term funding for SCC provided by AHS across Australia is necessary to appropriately reduce smoking rates. While the national Tackling Indigenous Smoking (TIS) program currently has smoking during pregnancy as a priority population, this health approach this does not cover all AHS across the country [46,55]. Therefore, there is a need for more activity-based funding to ensuring effective ongoing SCC provided by AHS. Allocation of Medicare item numbers for providing SCC could be considered with rigorous evaluation of the outcome, which has previously been advocated by experts [44,56].

## 5. Conclusions

In a cross-sectional online survey with 103 Aboriginal women, AHS was found to increase the chance of being smoke-free in pregnancy four fold. AHS are crucial to the implementation of health care provision to Aboriginal women who want to give up smoking in pregnancy. Continuation and improvement of this support is warranted with consistent allocation of funding. Pregnant Aboriginal women’s agency and efforts to protect their babies from tobacco-related harm need to be recognised. SCC will support Aboriginal women more meaningfully if more smoking cessation interventions are developed and implemented which take their quitting experiences into account. Our finding that urban participants reported being less likely to have smoke-free pregnancies conflicts with existing research and is worthy of further investigation.

## Figures and Tables

**Figure 1 ijerph-18-11240-f001:**
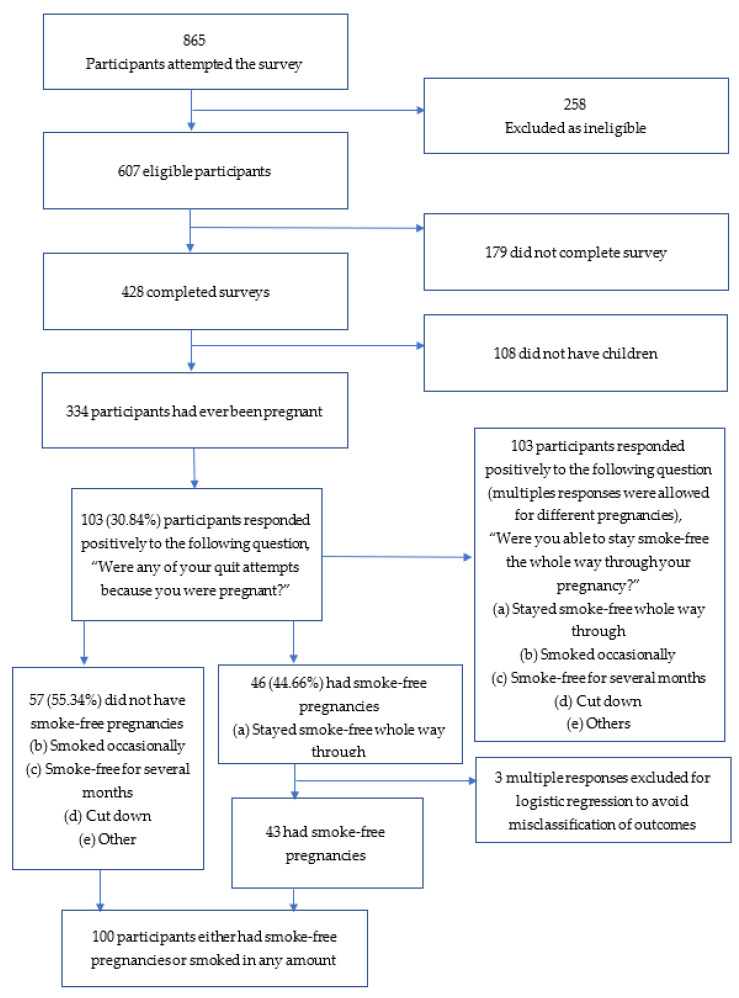
Diagram showing recruitment of participants.

**Table 1 ijerph-18-11240-t001:** Participant characteristics (N = 103).

Variables	Counts (Column %)
**Are you Aboriginal and/or Torres Strait Islander? ***	
Aboriginal	98 (95.15)
Aboriginal and Torres Strait Islander	5 (4.85)
**What is your age?**	
**Mean and Standard Deviation (sd)** (mean 32.13, sd ± 7.47)	
16–25	21 (20.39)
26–35	45 (43.69)
36 and above	37 (35.92)
**What education level have you completed?**	
Up to Year 11	35 (33.98)
Year 12	20 (19.42)
Current student at university/TAFE/Apprentice	20 (19.42)
Trade certificate/University degree	28 (27.18)
**What state do you live in?**	
NSW	47 (45.63)
QLD	29 (28.16)
VIC	12 (11.56)
Rest of Australia	15 (14.56)
**Do you use an Aboriginal Health Service(s)?**	
Yes	79 (76.70)
No	24 (23.30)
**What is your post code? (Remoteness)**	
Urban	58 (56.31)
Regional/remote	45 (43.69)
**How many children currently live in your household?**	
None	3 (2.91)
1–2	53 (51.46)
3 or more	44 (42.72)
**Pregnant**	6 (5.83)
**How long ago was your latest quit attempt**	
Days	11 (10.68)
Weeks	13 (12.62)
Months	79 (76.70)
**Of all the times you tried to quit smoking, what was the longest period you stayed completely of the smokes for?**	
Hours	2 (1.94)
Days	7 (6.80)
Weeks	13 (12.62)
Months	33 (32.04)
Years	46 (44.66)
Don’t know	2 (1.94)
**On your most recent quit attempt, did you stop smoking suddenly or did you gradually cut down your smokes?**	
Stopped suddenly	59 (57.28)
Reduced gradually	44 (42.72)
Have you ever used any type stop-smoking medications?	
Yes	40 (38.83)
No	63 (61.17)
**Would you say you are**	
Current smokers	59 (57.28)
Ex-smokers	44 (42.72)
**In the last month, have you tried to cut down the number of smokes you have each day? ****	
Yes	40 (67.80)
No	19 (32.20)
**Heaviness of Smoking Index ****	
Low	40 (67.80)
Moderate	18 (30.51)
High	1 (1.69)
**How much of the time have you felt the urge to smoke in the last 24 h? (FUTS) ****	
Not at all	5 (8.47)
A little of the time	5 (8.47)
Some of the time	17 (28.81)
A lot of the time	20 (33.90)
Almost all the time	4 (6.78)
All the time	8(13.56)
**FUTS (low vs. high) ****	
Low	27 (45.76)
High	32 (54.24
**In general how strong are your urges to smoke (in the last 24 h)? (SUTS) ***	
No urges	3 (5.08)
Slight	5 (8.47)
Moderate	21 (35.59)
Strong	17 (28.81)
Very strong	6 (10.17)
Extremely strong	7 (11.86)
**SUTS (low vs high) ****	
Low	29 (49.15)
High	30 (50.85)

* No Torres Strait Islander participants in this subsample. ** Current smokers only.

**Table 2 ijerph-18-11240-t002:** Participant characteristics of those who stayed entirely smoke-free during pregnancy and those who did not (N = 100).

Variables	Stayed Smoke-Free Whole Way through Pregnancy (n = 43)		Did Not Stay Smoke-Free Whole Way through Pregnancy (n = 57)		Pearson’s Chi-Square Test		
	n	Column %	n	Column % *	χ^2^	df	*p*-Value
**What is your age?**							
**Mean and Standard Deviation (sd)**	33.02 (7.52)		31.49 (7.67)				0.318
16–25	8	18.60	13	22.81			
26–35	17	39.53	25	43.86			
36 and above	18	41.86	19	33.33	0.80	2	0.671
**What education level have you completed?**							
Up to Year 11	15	34.88	20	35.09			
Year 12	7	16.28	13	22.81			
Current student at university/TAFE/Apprentice	8	18.60	11	19.30			
Trade certificate/University degree	13	30.23	13	22.81	1.05	3	0.790
**Do you use an Aboriginal Health Service(s)?**							
Yes	39	90.70	38	66.67			
No	4	9.30	19	33.33	7.99	1	0.005
**What is your post code? (Remoteness)**							
Urban	17	39.53	38	66.67			
Regional/Remote	26	60.47	19	33.33	7.29	1	0.007
**How long ago was your latest quit attempt ^∆^**							
Days	4	9.30	7	12.28			
Weeks	4	9.30	9	15.79			
Months	35	81.40	41	71.93			0.583
**Of all the times you tried to quit smoking, what was the longest period you stayed completely of the smokes for? ^∆^**							
Hours	0	0.00	2	3.51			
Days	1	2.33	6	10.53			
Weeks	0	0.00	13	22.81			
Months	9	20.93	23	40.35			
Years	32	74.42	12	21.05			
Don’t know	1	2.33	1	2.33			<0.001
**On your most recent quit attempt, did you stop smoking suddenly or did you gradually cut down your smokes?**							
Stopped suddenly	27	62.79	27	47.37			
Reduced gradually	16	37.21	30	52.63	1.03	1	0.310
**Have you ever used any type stop-smoking medications?**							
Yes	13	30.23	24	42.11			
No	30	69.77	33	57.89	1.48	1	0.223
**In the last month, have you tried to cut down the number of** **smokes you have each day? *^,∆^**							
Yes	9	69.23	31	67.39			
No	4	30.77	15	32.61			1.000
**How many cigarettes do you usually smoke per day (on the days you smoke)? ^†^**							
**Number of cigarettes smoked per day; Median and (Q1–Q3)**	6 (4–10)		10 (6–15)				0.134
**Heaviness of Smoking Index *^,∆^**							
Low	11	84.62	29	63.04			
Moderate	2	15.38	16	34.78			
High	0	0.00	1	2.17			0.460
**How much of the time have you felt the urge to smoke in the last 24 h? (FUTS) ***							
Low	8	61.54	19	41.30			
High	5	38.46	27	58.70	1.67	1	0.196
**In general how strong are your urges to smoke (in the last 24 h)? (SUTS) ***							
Low	8	61.54	21	45.65			
High	5	38.46	25	54.35	1.02	1	0.312

* Current smokers only. ^†^ Mann–Whitney U-test. ^∆^ Fisher’s exact test was performed where cell frequency was <5.

**Table 3 ijerph-18-11240-t003:** Univariate and multivariable logistic regression for identifying factors associated with smoke-free pregnancy (N = 100).

Variables			Multivariable Analysis		
	n	%	AOR ^†^	95% CI ^∆^ for OR	*p*-Value
**What is your age?**					
16–25	21	21	Ref		
26–35	42	42	1.73	0.46–6.48	0.418
36 and above	37	34	2.10	0.55–8.09	0.280
**What education level have you completed?**					
Up to Year 11	35	35	Ref		
Year 12	20	20	0.85	0.23–3.07	0.800
Current student at university/TAFE/Appr	19	19	1.76	0.47–6.56	0.401
Trade certificate/University degree	26	26	1.45	0.46–4.58	0.530
**Have you ever used any type stop-smoking medications?**					
No	63	63	Ref		
Yes	37	37	0.56	0.21–1.47	0.235
**Do you use an Aboriginal Health Service(s)?**					
No	23	23	Ref		
Yes	77	73	4.54	1.29–15.95	0.018
**What is your post code? (Remoteness)**					
Regional/remote	45	45	Ref		
Urban	55	55	0.33	0.13–0.84	0.020

^†^ AOR = Adjusted odds ratio; Adjusted for age, education, and previous use of smoking cessation medications. ^∆^ CI = Confidence interval.

**Table 4 ijerph-18-11240-t004:** Major themes around changes in smoking behaviour in the context of pregnancy.

Sl	Major Themes	Subthemes	Counts
1	Motivations for changes in the smoking behaviour	Quitting soon after becoming aware of pregnancy with strong intent to protect unborn child	9
		Health of unborn child	17
		Morning sickness making smoking not pleasurable and even difficult	16
2	The level of behavioural change may vary	Reduction in consumption	18
		Complete cessation through to term	28
3	The experience of change is diverse, either difficult or easy	Giving up smoking can be either difficult or stressful for some Aboriginal women during pregnancy	34
		Some Aboriginal women may find quitting smoking easy during pregnancy	18
4	Experience of change evolves over time and in different pregnancies	Change in smoking behaviour over the course of a pregnancy	13
		Smoking and quitting experience may vary in different pregnancies	12

## Data Availability

In line with the Aboriginal Health and Medical Research Council (AH&MRC), the data is non-sharable.

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
