# Peer review of "Factors Associated with Smoke-Free Pregnancy among Aboriginal and Torres Strait Women and Their Experience of Quitting Smoking in Pregnancy: A Mixed Method Cross-Sectional Study"

_ijerph, 2021, doi:10.3390/ijerph182111240_

Round 1
Reviewer 1 Report
This is a wonderful paper. Your design and analysis were robust and create a wonderful picture of the situation.
How was the sample size determined? Did you conduct a power analysis? What is the smoking prevalence of the communities generally? Meaning is smoking among Aboriginal/Torres Strait women of childbearing age proportionate to the general population? Could the general population smoking rate be an influence?
Author Response
Response to reviewers’ comments
Manuscript ID ijerph-1377994 entitled " Factors associated with smoke-free pregnancy among Aboriginal and Torres Strait women and their experience of quitting smoking in pregnancy: A mixed-method cross-sectional study "
We thank the reviewer for their constructive comments. We believe the comments have improved our work. Please find our response to the individual comments below. As we addressed the comments the page numbers and line numbers changed. Therefore, we mentioned page number corresponding to all changes. All changes are in track changes in the manuscript.
Reviewer: 1
Comments to the Author
This is a wonderful paper. Your design and analysis were robust and create a wonderful picture of the situation.
How was the sample size determined?
RESPONSE:
ADDED:
“Sample size was informed by Ogundimu et al. (2016) [Ref] and calculated based on a minimum of ≥10 events per variable. Given the variable of interest was quitting during pregnancy, a minimum sample size of 84 Aboriginal and Torres Strait women was required (i.e., 12% of expecting Aboriginal and Torres Strait women quit smoking during pregnancy, [Ref] and therefore to reach ≥10 events per variable, we required a minimum of 84 participants). The survey ended for potential participants who were deemed ineligible based on their responses to the screening questions (Figure 1).” (p4, l150-157)
- Did you conduct a power analysis?
RESPONSE:
A POWER ANALYSIS WAS CONDUCTED FOR THE PARENT STUDY. A MINIMUM 385 WOMEN WERE REQUIRED FOR A 5% MARGIN OF ERROR. WHILE WE HAVE NOT INCLUDED THIS DETAIL IN THE MANUSCRIPT, WE HAVE NOW REFINED THE SAMPLING DETAIL IN THE MANUSCRIPT. (p4,l150-157)
- What is the smoking prevalence of the communities generally? Meaning is smoking among Aboriginal/Torres Strait women of childbearing age proportionate to the general population?
RESPONSE:
THE AUTHORS HAVE NOW ADDED THIS INFORMATIONIN THE INTRODUCTION. (p2, l61)
- Could the general population smoking rate be an influence?
RESPONSE:
THANK YOU FOR YOUR COMMENT AND THE AUTHORS AGREE THAT TOBACCO CONTROL POLICIES IN AUSTRALIA COULD INFLUENCE QUIT ATTEMPTS AMONG ABORIGINAL AND TORRES STRAIT ISLANDER PEOPLE. HOWEVER, THIS STUDY REPORTED ONLY ABORIGINAL WOMEN’S EXPERIENCES AND THERE IS NO EVIDENCE TO LINK GENERAL POPULATION INTERVENTIONS WITH THESE FINDINGS.

Reviewer 2 Report
The present study is an important contribution to the literature and reports a mixed-methods results on factors associated with smoke-free pregnancy and quitting smoking in pregnancy among Aboriginal and Torres Strait women. However, the paper can be improved.
Introduction
- Lines 38-44 would be better suited at the end of the Introduction section. This section could also be improved by highlighting the rare nature of this study given the limited literature on quitting smoking among Aboriginal women and First Nations peoples
- The Introduction would be improved by providing additional information about the prevalence of smoking among Aboriginal and Torres Strait Islander women overall and specifically among pregnant women. Also provide the prior smoking prevalence among pregnant Aboriginal women from the past decade.
- The authors generally acknowledge that smoking in pregnancy is a major public health concern among First Nation peoples throughout high income countries and that colonization has contributed to health inequities among this community. The authors also end the first paragraph by saying that the present study seeks to examine the progress in Aboriginal women’s health but they do not provide enough context about the prior state of Aboriginal women’s health or smoking in pregnancy. The Introduction would be further improved by expanding the points already and providing additional context on how smoking negatively impacts the health of aboriginal women and the factors that contribute to smoking/smoking in pregnancy among aboriginal women. Additionally, consider framing the Introduction around the study’s results. While colonization and oppression is a contributor to smoking and poor health among socially disadvantaged groups, it did not come up explicitly in the qualitative results nor was discrimination/oppression assessed quantitatively. As such, it is relevant and appropriate to mention these issues in the Introduction, but just make sure that it is not the sole/main framing of this section.
- Line 59/ 60 may be unclear for readers, especially the use of privileging and gains. Please consider re-wording.
- In line 64 the authors state that “leveraging the factors that enable smoke-free pregnancies…” but the introduction does not clearly or adequately present factors that enable smoke-free pregnancies overall or among Aboriginal women. Please re-organize and expand the introduction to more clearly outline barriers and facilitators to smoke-free pregnancies among Aboriginal women.
- In the last paragraph the authors also state that evidence in the present study may also be relevant to addressing smoking during pregnancy among First Nation women in other high-income countries. Although this may be true, as it is currently written, the Introduction presents no evidence to support this claim. The Introduction needs to either be better set-up with information comparing risk factors for smoking during pregnancy among First Nation women across high-income countries, the sentence needs to be removed, or the sentence can be relocated to the Discussion section provided that adequate evidence is provided to support the sentence.
Materials and Methods
- Please add additional information on what the AHS is and its relevance to the Aboriginal community’s health in line 90. Presently, additional context is only provided in the Discussion and readers outside of NSW/New Zealand are unable to understand the relevance of your findings regarding the participant’s use of AHS.
- Based on the Sampling section, the snowball sampling was used and the sampling frame and generalizability of the study are not clear. As such, it is not appropriate to refer to the study as a national study (lines 73 and 474).
- In ‘Sampling’ section, add what happened after prospective participants were deemed ineligible.
- Report how many people were screened, deemed eligible/ineligible, completed the survey, etc. and reference Figure 1 sooner.
- In the ‘Data Analysis’ section, define the analytic sample and clarify whether analyses were only restricted to participants who reported having any children living in the home/were currently pregnant and then stratified by smoke-free pregnancy status. For example, if women who do not have children in the home were included in the analyses this would be incorrect and it needs to be made clear to readers which participants the sample size is based on.
- Please acknowledge the measures and assessment of smoke-free pregnancy status as a limitation. Based on Table S1 it appears that motherhood was based on having a child living within in the home or being currently pregnant. Although I am unfamiliar with Aboriginal family structure and dynamics, I would assume that it may be possible for women to have previously had a child who is not currently living in their home whether due to age, the child living with another guardian, health reasons, premature death, etc. Similarly, women could possibly have children living in the home that they did not give birth to.
- Clarify whether smoke-free pregnancy status was based on having at least 1 smoke-free pregnancy
- Report the number of participants with multiple pregnancies who were excluded from analyses for reporting a smoke-free and non-smoke free pregnancy
- Clarify when a t test or Mann Whitney U test was used
- Report the alpha level used to determine statistical significance
- Report Inter-coder/inter-rater reliability and how any discrepancies were resolved. Also clarify whether TR conducted final coding of the qualitative data or was the only person to formally analyze the qualitative data. If it is the latter, then this should be acknowledged as a limitation.
Results
- Figure 1 is inappropriately referenced first in the results section. Several revisions suggested above could have been addressed if Figure 1 was referenced earlier. Otherwise, readers will be left without this information while reading the materials/methods section.
- Text is cut off in the 3rd box from the bottom in Figure 1. Please revise. Also clarify why 3 participants/multiple responses were excluded from regression analyses.
- Define TAFE before using acronym
- In the methods section clarify whether age was collected only as a categorical variable or if it was collected as a continuous variable and then later categorized for analyses. Based on Table 1 and Table S1 it is confusing on how mean age was calculated when only categories are presented in Table S1.
- Clarify whether the 3 respondents who reported having no children in the home were excluded from analyses. Based on Table S1/skip patterns it appears that these participants should be excluded as their smoke-free pregnancy status would not have been assessed.
- Provide explanation of the State acronyms. IJERPH is an international journal and not all readers will be familiar.
- If possible, report the average or median time since last quit attempt and longest period completely smoke-free
- Add foot-notes to clarify the meaning of the HSI, FUTS, and SUTS scores
- Text in lines 190-192 would be better suited above (see comments in Materials/methods section).
- Remove vertical lines from Tables 1 and 2. Consider APA formatting of tables similar to Table 3. Also consider adding boldface to statistically significant results throughout and just note that column %s are reported in Table 2 instead of noting it with **. These suggestions will improve readability and interpretability of the superscripts.
- Move the † superscript for a Mann Whitney U test next to the variable name like all other superscripts were placed instead of next to the p-value
- Reporting of the univariate results in Table 3 are redundant with the results in Table 2 except the specific pairwise differences are reported as ORs. Remove the univariate regression results and only report the multivariate results.
- Widen the first column/variable labels in Table 3 so that they align properly with the results
- Add footnote to Table 3 to denote what the analyses adjusted for.
- Revise line 212 to state that analyses adjusted for these variables
- Line 265 consider using a synonym to nil for more universal interpretability
- Line 268/269 seems to have an extra paragraph break
Discussion
- The authors should be careful not to overstate their findings in the discussion starting in lines 356 and 380. Based on Table S1, the quantitative measures used in the present study did not assess the any time-frame when smoking cessation attempts were made during the pregnancy. Thus, the comparability of these results to the national smoking cessation rate during pregnancy is limited and the imprecision of this measure should be acknowledged as a limitation. Although the authors frame the assessment of quit attempts beyond 20 weeks of gestation as a strength (starting line 371), they should also be mindful that they are limited in their own assessment of what stage participants quit/reduced smoking during gestation.
- Similarly, the discussion/implications starting in line 393 is less explicitly supported by the findings of this study (although this is likely true at least anecdotally). Moreover, the qualitative data used to support these claims may not be as rigorous as other forms of qualitative data collection including focus groups, in-depth interviews, etc. The authors must better link these points to the actual findings of the present study and framing used in the Introduction/study rationale or limit/remove these implications from the paper
- In the same vein, the authors should be careful not to overstate the implications of their findings in lines 443-447. The data are cross-sectional based on a convenience sample and with a small sample size so the generalizability of the findings and their rigor are limited.
- The implications/policy implications related to AHS are sound and appropriate based on the results. The Discussion of the AHS-related results can be further expanded in the paragraph starting in line 343
- Conclusion should be revised accordingly based on other revisions.
Table S1
- In the last column of #10 (pg 3), there appears to be a typo and the Q1. Number of cigarettes smoked per day is incorrectly repeated. Please revise as needed and check throughout for other errors (for example, the word ‘of’ may be missing in #21).
- The HSI scoring for time to first cigarette after waking are incorrect. Having a shorter time to first cigarette (i.e., smoking first cigarette within 5 minutes of waking) indicates greater nicotine dependence and thus, a greater HSI score. If HSI was calculated using the scoring in Table S1 it is incorrect and will need to be recalculated and any results related to HIS would need to be reinterpreted. Please correct the HSI scoring in Table S1 and the analyses/results if necessary.
- Add a title/HSI score to #11
Table S2
- Clarify the meaning of the highlights or remove
Author Response
Response to reviewers’ comments
Manuscript ID ijerph-1377994 entitled " Factors associated with smoke-free pregnancy among Aboriginal and Torres Strait women and their experience of quitting smoking in pregnancy: A mixed-method cross-sectional study "
We thank the reviewer for their constructive comments. We believe the comments have improved our work. Please find our response to the individual comments below. As we addressed the comments the page numbers and line numbers changed. Therefore, we mentioned page number corresponding to all changes. All changes are in track changes in the manuscript.
Reviewer: 2
Comments to the Author
The present study is an important contribution to the literature and reports a mixed-methods results on factors associated with smoke-free pregnancy and quitting smoking in pregnancy among Aboriginal and Torres Strait women. However, the paper can be improved.
- Lines 38-44 would be better suited at the end of the Introduction section. This section could also be improved by highlighting the rare nature of this study given the limited literature on quitting smoking among Aboriginal women and First Nations peoples.
RESPONSE:
LINE 38-44 HAS BEEN MOVED AT THE END OF THE INTRODUCTION SECTION.
THE FOLLOWING LINES HAS BEEN ADDED AS ADVISED BY THE REVIEWER.
“The existing literature presents evidence on Aboriginal and Torres Strait Islander women’s experience of smoking cessation with a focus their knowledge and perception regarding harms of smoking during pregnancy [Ref]. However, they offer little insight into what helps Aboriginal and Torres Strait Islander women to quit smoking during pregnancy and their quitting behaviour.” (p2, l83-92)
- The Introduction would be improved by providing additional information about the prevalence of smoking among Aboriginal and Torres Strait Islander women overall and specifically among pregnant
women. Also provide the prior smoking prevalence among pregnant Aboriginal women from the past decade.
RESPONSE:
THE FOLLOWING LINES HAS BEEN ADDED AS ADVISED BY THE REVIEWER.
“Although, 42% of Aboriginal and Torres Strait Islander women aged 18 or more smoke tobacco in any amount [Ref], over the past decade, the proportion of Aboriginal women who smoked during pregnancy declined by approximately eight percentage points (44% in 2017 vs 52% in 2009).” (p2, l58-61)
- The authors generally acknowledge that smoking in pregnancy is a major public health concern among First Nation peoples throughout high income countries and that colonization has contributed to health inequities among this community. The authors also end the first paragraph by saying that the present study seeks to examine the progress in Aboriginal women’s health but they do not provide enough context about the prior state of Aboriginal women’s health or smoking in pregnancy. The Introduction would be further improved by expanding the points already and providing additional context on how smoking negatively impacts the health of aboriginal women and the factors that contribute to smoking/smoking in pregnancy among aboriginal women. Additionally, consider framing the Introduction around the study’s results. While colonization and oppression is a contributor to smoking and poor health among socially disadvantaged groups, it did not come up explicitly in the qualitative results nor was discrimination/oppression assessed quantitatively. As such, it is relevant and appropriate to mention these issues in the Introduction, but just make sure that it is not the sole/main framing of this section.
RESPONSE:
THE INTRODUCTION SECTION HAS BEEN CHANGED BY INCLUDING MORE LITERATURE TO SET THE CONTEXT OF THE PAPER, AS ADVISED BY REVIEWER 2.
“Smoking during pregnancy is more common among First Nation women compared to women in the general population in high-income countries [Ref]. Social disadvantage, stressful life events, lack of access to culturally appropriate smoking cessation support, and high prevalence of smoking in First Nation communities, owing to the legacy of colonisation instilled into public policies, are some of the major contributing factors to persisting smoking during pregnancy [Ref]. First Nation women are, thus, at higher risk of smoking-related adverse pregnancy outcomes including placenta previa, preterm birth, intrauterine growth restriction, spontaneous abortion, and stillbirth [Ref]. It also elevates the risk of low birth weight, respiratory problems and related hospitalisation, and behavioural problems in children, and risks of obesity and hypertension in their early adulthood [Ref]. However, quitting smoking because of pregnancy was found to be common among First Nation women [Ref].” (p1-2, l37-48)
- Line 59/ 60 may be unclear for readers, especially the use of privileging and gains. Please consider re-wording.
RESPONSE:
CHANGED TO ‘highlighting’ and ‘progress’ (p2, l75-76)
- In line 64 the authors state that “leveraging the factors that enable smoke-free pregnancies…” but the introduction does not clearly or adequately present factors that enable smoke-free pregnancies overall or among Aboriginal women. Please re-organize and expand the introduction to more clearly outline barriers and facilitators to smoke-free pregnancies among Aboriginal women.
RESPONSE:
THE INTRODUCTION SECTION HAS BEEN CHANGED BY INCLUDING MORE LITERATURE TO SET THE CONTEXT OF THE PAPER AND INCLUDING EXISTING EVIDENCE ON BARRIER AND ENABLERS OF SMOKING CESSATION DUIRNG PREGNANCY.
ADDED:
“On the contrary, high motivation to give up smoking, resilience and support from family and community are major enablers to smoking cessation among Aboriginal women [Ref].” (p2, l64-66)
“The existing literature presents evidence on Aboriginal and Torres Strait Islander women’s (hereafter ‘Aboriginal women’, with acknowledgement of autonomy of all Aboriginal and Torres Strait Islander peoples) experience of smoking cessation with a focus their knowledge and perception regarding harms of smoking during pregnancy [Ref]. However, they offer little insight into what helps Aboriginal women to quit smoking during pregnancy and their quitting behaviour. This study was conducted by an Aboriginal-led research team in partnership with Aboriginal communities in urban and regional New South Wales (NSW), Australia. It seeks to examine the progress that has been taking place in Aboriginal women’s health by examining factors associated with smoke-free pregnancy.” (p2, l83-92)
- In the last paragraph the authors also state that evidence in the present study may also be relevant to addressing smoking during pregnancy among First Nation women in other high-income countries. Although this may be true, as it is currently written, the Introduction presents no evidence to support this claim. The Introduction needs to either be better set-up with information comparing risk factors for smoking during pregnancy among First Nation women across high-income countries, the sentence needs to be removed, or the sentence can be relocated to the Discussion section provided that adequate evidence is provided to support the sentence.
RESPONSE:
THE CHANGES IN THE INTRODUCTION NOW PROVIDE MORE CONTEXT TO THIS STATEMENT.
- Please add additional information on what the AHS is and its relevance to the Aboriginal community’s health in line 90. Presently, additional context is only provided in the Discussion and readers outside of NSW/New Zealand are unable to understand the relevance of your findings regarding the participant’s use of AHS.
RESPONSE:
ADDED:
“AHS or Aboriginal Community Controlled Health Services are the main provider of Aboriginal primary health care in Australia. AHS are key players in ensuring improvement in Aboriginal health and wellbeing and developing health Aboriginal communities. Clinical service, health promotion, cultural safety and community engagement are some major components of the distractive model of care of AHS [Ref].” (p3, l119-124)
- Based on the Sampling section, the snowball sampling was used and the sampling frame and generalizability of the study are not clear. As such, it is not appropriate to refer to the study as a national study (lines 73 and 474).
RESPONSE:
THE HAVE NOW REFINED THE SAMPLING SECTION AND PROVIDED MORE DETAIL. THE WORD ‘NATIONAL’ HAS NOW BEEN DELETED.
- In ‘Sampling’ section, add what happened after prospective participants were deemed ineligible.
RESPONSE:
ADDED:
“The survey ended for potential participants who were deemed ineligible based on their responses to the screening questions.” (p4, l156 -157)
- Report how many people were screened, deemed eligible/ineligible, completed the survey, etc. and reference Figure 1 sooner.
RESPONSE:
WE HAVE NOW MOVED FIGURE 1 CLOSER TO SAMPLING. BEING MINDFUL OF THE LENGTH OF THE PAPER WE DID NOT REPEAT THE INFORMATION ALREADY GIVEN IN FIGURE 1.
- In the ‘Data Analysis’ section, define the analytic sample and clarify whether analyses were only restricted to participants who reported having any children living in the home/were currently pregnant and then stratified by smoke-free pregnancy status. For example, if women who do not have children in the home were included in the analyses this would be incorrect and it needs to be made clear to readers which participants the sample size is based on.
RESPONSE:
THIS HAS NOW BEEN CLARIFIED.
ADDED:
“This analysis compares Which Way? participants who reported having at least one a smoke-free pregnancy versus those who reported not having a smoke-free pregnancy. However, participants with multiple pregnancies who reported being both smoke-free in one pregnancy and continued to smoke in another were excluded from tests of associating and logistic regression models to aid the analysis and avoid misclassification of outcomes.” (p6, l257-261)
“Participants were included in the analysis independent of the number of children living in their house or current pregnancy.” (p6, l274-275)
- Please acknowledge the measures and assessment of smoke-free pregnancy status as a limitation. Based on Table S1 it appears that motherhood was based on having a child living within in the home or being currently pregnant. Although I am unfamiliar with Aboriginal family structure and dynamics, I would assume that it may be possible for women to have previously had a child who is not currently living in their home whether due to age, the child living with another guardian, health reasons, premature death, etc. Similarly, women could possibly have children living in the home that they did not give birth to.
RESPONSE:
WE WOULD LIKE TO CLARIFY THAT WE DID NOT MEASURE PREGNANCY OR MOTHERHOOD BY THE NUMBER OF CHILDREN LIVING IN THE HOUSE IN THIS ANALYSIS. WE DETERMINE SMOKE-FREE PREGNANCY, WHICH IS OUR MAIN OUTCOME, BY ASKING THE FOLLOWING QUESTIONS:
- I) WERE ANY OF YOUR QUIT ATTEMPTS BECAUSE YOU WERE PREGNANT?
THOSE WHO REPLIED IN POSITIVE WERE THEN ASKED:
- II) WERE YOU ABLE TO STAY SMOKE-FREE THE WHOLE WAY THROUGH YOUR PREGNANCY?
(A) STAYED SMOKE-FREE WHOLE WAY THROUGH;
(B) SMOKED OCCASIONALLY;
(C) STAYED SMOKE-FREE FOR SEVERAL MONTHS;
(D) CUT DOWN; AND
(E) OTHERS.
THIS POINT HAS NOW BEEN CLARIFIED FURTHER. PLEASE SEE RESPONSE TO THE PREVIOUS COMMENTS.
- Clarify whether smoke-free pregnancy status was based on having at least 1 smoke-free pregnancy.
RESPONSE:
THIS POINT HAS NOW BEEN CLARIFIED FURTHER. PLEASE SEE RESPONSE TO THE PREVIOUS COMMENTS.
- Report the number of participants with multiple pregnancies who were excluded from analyses for reporting a smoke-free and non-smoke free pregnancy.
RESPONSE:
THE NUMBER OF PARTICIPANTS EXCLUDED FROM THE LOGISTIC REGRESSION ANALYSIS HAS ALREADY BEEN MENTIONED (p11, l431 OF THE CURRENT VERSION).
- Clarify when a t test or Mann Whitney U test was used
RESPONSE:
THIS HAS NOW BEEN CLARIFIED.
“Differences in means in continuous variables between these groups were measured by performing student’s t-tests and Mann Whitney U-tests depending on the distribution of the data and number of observations. Student’s t-test was used to estimate mean age of the participants while Mann Whitney U-test was used to estimate the mean number of cigarettes smoked daily by those who identified as smokers.” (p5, l289-294)
- Report the alpha level used to determine statistical significance
RESPONSE:
ALFA LEVEL HAS NOW BEEN ADDED.
“Alpha level was set at 5%.” (p6, I294)
- Report Inter-coder/inter-rater reliability and how any discrepancies were resolved. Also clarify whether TR conducted final coding of the qualitative data or was the only person to formally analyze the qualitative data. If it is the latter, then this should be acknowledged as a limitation.
RESPONSE:
TR WAS NOT THE ONLY PERSON WHO FORMALLY ANALYSED THE QUALITATIVE DATA. THIS CONCERN HAS BEEN CLARIFIED BY ADDING THE FOLLOWING TEXT:
“Any discrepancies in coding performed by TR and MK were resolved through discussion. TR performed a higher order thematic analysis with frequent reflexive discussion with MK.” (p7, l314-316)
- Figure 1 is inappropriately referenced first in the results section. Several revisions suggested above could have been addressed if Figure 1 was referenced earlier. Otherwise, readers will be left without this information while reading the materials/methods section.
RESPONSE:
FIGURE 1 HAS NOW BEEN MOVED CLOSER TO THE SAMPLING SECTION. (p5)
- Text is cut off in the 3rdbox from the bottom in Figure 1. Please revise. Also clarify why 3 participants/multiple responses were excluded from regression analyses.
RESPONSE:
THIS HAS NOW BEEN CORRECTED.
- Define TAFE before using acronym
RESPONSE:
THE FULL FORM OF TAFE HAS NOW BEEN ADDED (p9, l420).
- In the methods section clarify whether age was collected only as a categorical variable or if it was collected as a continuous variable and then later categorized for analyses. Based on Table 1 and Table S1 it is confusing on how mean age was calculated when only categories are presented in Table S1.
RESPONSE:
THIS HAS NOW BEEN INCLUDED. (p6, l290).
- Clarify whether the 3 respondents who reported having no children in the home were excluded from analyses. Based on Table S1/skip patterns it appears that these participants should be excluded as their smoke-free pregnancy status would not have been assessed.
RESPONSE:
AS MENTIONED IN RESPONSE TO COMMENT 12, SMOKE-FREE PREGNANCY WAS NOT DETERMINED BY NUMBER OF CHILDREN LIVING IN THE HOUSE.
- Provide explanation of the State acronyms. IJERPH is an international journal and not all readers will be familiar.
RESPONSE:
STATE NAMES HAVE BEEN MENTIONED IN FULL.
- If possible, report the average or median time since last quit attempt and longest period completely smoke-free
RESPONSE:
DATA ON LONGEST QUIT ATTEMPTS AND TIME SINCE LAST QUIT ATTEMPTS WAS COLLECTED AS A CATEGORICAL VARIABLE WHERE THE CATEGORIES WERE THE FOLLOWING:
LONGEST QUIT ATTEMPTS
- A) HOURS
- B) DAYS
- C) WEEKS
- D) MONTHS
- E) YEARS
- F) DON’T KNOW
TIME SINCE LAST QUIT ATTEMPTS
- A) DAYS
- B) WEEKS
- C) MONTHS
- D) NEVER
THEREFORE, IT IS NOT FEASIBLE TO ESTIMATE THE MEAN OR MEDIAN.
- Add foot-notes to clarify the meaning of the HSI, FUTS, and SUTS scores
RESPONSE:
THANK YOU FOR THIS COMMENT. WHILE ADDING A FOOTNOTE MIGHT GUIDE THE READER UNFAMILIAR WITH TOBACCO INDICATORS, GIVEN THE COMPLEXITY OF THE SCORING IT WAS DEEMED MORE APPROPRIATE IN A SUPPLEMENTARY FILE. HOWEVER, IN THE SUPPLEMENTARY FILE WE HAVE NOW EXPLAINED THE SCORING OF HSI, FUTS, AND SUTS.
“The HSI consists of two questions: a) “How many cigarettes do you usually smoke per day (on the days you smoke)?”, and b) “On the morning that you have a smoke, how soon do you have it after waking up?” The response options for the latter question are (1) More than 60 minutes; (2) 31-60 minutes; (3) 6-30 minutes; and (4) Within 5 minutes. The HSI score ranges between 0 and 6. This score is then classified into three dependence categories: low dependence (scores between 0 and 2); moderate dependence (scores of 3 and 4) and high dependence (scores of 5 and 6). Data were also collected on frequency of urges to smoke (FUTS) (in the last 24 hours) and strength of urges to smoke (SUTS), both have been previously used with Aboriginal communities. FUTS was measured by a single question: “How much of the time have you felt the urge to smoke in the last 24 h?”. The response options are Not at all (0); A little of the time; (1) Some of the time; (2) A lot of the time (3); Almost all the time (4); and All the time (5). Participants were grouped into two categories; those scoring between 0-2 were categorized as low FUTS and those who scored between 3-5 were considered high FUTS. Likewise, SUTS was measured by a single question: “In general, how strong are your urges to smoke (in the last 24 h)?”. Scores were categories as low (0-2) and high (3-5).”
- Text in lines 190-192 would be better suited above (see comments in Materials/methods section).
RESPONSE:
THE ABOVE LINE NUMBERS WERE FOUND WITHOUT ANY CORRESPONDING TEXT. HOWEVER, ALL CHANGES ADVISED IN THE MATERIALS/METHODS SECTION HAVE BEEN ADDRESSED.
- Remove vertical lines from Tables 1 and 2. Consider APA formatting of tables similar to Table 3. Also consider adding boldface to statistically significant results throughout and just note that column %s are reported in Table 2 instead of noting it with **. These suggestions will improve readability and interpretability of the superscripts.
RESPONSE:
THE TABLES HAVE NOW BEEN CORRECTED FOLLOWING REVIEWERS’ ADVICE.
- Move the †superscript for a Mann Whitney U test next to the variable name like all other superscripts were placed instead of next to the p-value
RESPONSE:
THESE SYMBOLS HAVE NOW BEEN CORRECTED FOLLOWING REVIEWERS’ ADVICE.
- Reporting of the univariate results in Table 3 are redundant with the results in Table 2 except the specific pairwise differences are reported as ORs. Remove the univariate regression results and only report the multivariate results.
RESPONSE:
UNIVARIATE REGRESSION RESULTS HAVE NOW BEEN REMOVED.
- Widen the first column/variable labels in Table 3 so that they align properly with the results
RESPONSE:
THIS ERROR HAS NOW BEEN CORRECTED.
- Add footnote to Table 3 to denote what the analyses adjusted for
RESPONSE:
FOOTNOTE HAS BEEN ADDED UNDERNEATH TABLE 3.
- Revise line 212 to state that analyses adjusted for these variables
RESPONSE:
REVISED (p13, I458).
- Line 265 consider using a synonym to nil for more universal interpretability
RESPONSE:
THIS HAS NOW BEEN CHANGED TO ‘AND QUIT’. (p15,I515).
- Line 268/269 seems to have an extra paragraph break
RESPONSE:
REVIEW.
- The authors should be careful not to overstate their findings in the discussion starting in lines 356 and 380. Based on Table S1, the quantitative measures used in the present study did not assess the any time-frame when smoking cessation attempts were made during the pregnancy. Thus, the comparability of these results to the national smoking cessation rate during pregnancy is limited and the imprecision of this measure should be acknowledged as a limitation. Although the authors frame the assessment of quit attempts beyond 20 weeks of gestation as a strength (starting line 371), they should also be mindful that they are limited in their own assessment of what stage participants quit/reduced smoking during gestation.
RESPONSE:
ADDED:
“About 30.84% of the participants in the Which Way? cohort attempted to give up smoking at some stage during their pregnancy (Figure 1). While this study did not report quitting at specific time points, our findings exceeds the national smoking cessation rate (22%) during pregnancy, in general [Ref].” (p17, l613-617)
“The qualitative evidence offers important insight into Aboriginal women’s experiences of giving up smoking in pregnancy. This is the first study to offer qualitative evidence on Aboriginal women’s quitting behaviour across different pregnancies, drawn from over a hundred women, and across geographical settings.” (p17, l625-629)
- Similarly, the discussion/implications starting in line 393 is less explicitly supported by the findings of this study (although this is likely true at least anecdotally). Moreover, the qualitative data used to support these claims may not be as rigorous as other forms of qualitative data collection including focus groups, in-depth interviews, etc. The authors must better link these points to the actual findings of the present study and framing used in the Introduction/study rationale or limit/remove these implications from the paper
RESPONSE:
WE THANK THE REVIEWER FOR THEIR COMMENT. WE NOTE COVID IMPACTS AND WHILE MORE RIGOROUS QUAL METHODS COULD BE USEFUL, NO QUALITATIVE STUDY HAS EVER CAPTURE SMOKING CESSATION EVIDENCE FROM SUCH A LARGE NUMBER OF ABORIGINAL WOMEN NUMBER OR GEOGRAPHICAL SCOPE OF THIS PROJECT. THE QUALITATIVE EVIDENCE OFFERS INSIGHT INTO ABORIGINAL WOMEN’S QUITTING EXPERIENCE, AND REINFORCES QUANTITATIVE FINDINGS OF SMOKING CESSATION.
- In the same vein, the authors should be careful not to overstate the implications of their findings in lines 443-447. The data are cross-sectional based on a convenience sample and with a small sample size, so the generalizability of the findings and their rigor are limited.
RESPONSE:
WHILE THE SAMPLE SIZE MAY BE CONSIDER SMALL FOR A GENERAL POPULATION RESEARCH. THIS STUDY WAS SPECIFIC TO ABORIGINAL AND TORRES STRAIT ISLANDER PEOPLE WHO ARE 3.3% OF THE POPULATION AS A WHOLE. HOWEVER THIS STUDY THEN ONLY FOCUSSED ON WOMEN OF REPRODUCTIVE AGE WHICH IS EQUIVALENT TO LESS THAN 1% OF THE AUSTRALIAN POPULATION.
ADDED:
“This is not a population representative sample. However, this study offers important evidence on Aboriginal women of reproductive age group who represent less than 1% of the Australian population [Ref]. Previous nationally representative studies such as Mayi Kuwayu report a 2.3% response rate with the highest respondents being over 50 years of age [Ref]. Acknowledging the challenges in engaging Aboriginal women of reproductive age in research, this study provides novel findings from an sufficient sample relevant to the main outcomes.” (p19, l712-718)
- The implications/policy implications related to AHS are sound and appropriate based on the results. The Discussion of the AHS-related results can be further expanded in the paragraph starting in line 343
RESPONSE:
WE THANK THE REVIEWER FOR THE COMMENT. WHILE WE WOULD LIKE TO EXPAND THE DISCUSSION ON AHS, WE ARE CONSIDERATE OF THE LENGTH OF THE PAPER. WE ALSO BELIEVE THAT THE CURRENT EXTENT OF THE DISCUSSION IS APPROPRIATE.
- Conclusion should be revised accordingly based on other revisions.
RESPONSE:
WE HAVE REVIEWED THE CONCLUSION AND WE BELIEVE THAT THIS SECTION IS COMPREHENSIVE AND IN LINE WITH OUR FINDINGS.
- In the last column of #10 (pg 3), there appears to be a typo and the Q1. Number of cigarettes smoked per day is incorrectly repeated. Please revise as needed and check throughout for other errors (for example, the word ‘of’ may be missing in #21).
RESPONSE:
THIS ERROR HAS NOW BEEN CORRECTED.
- The HSI scoring for time to first cigarette after waking are incorrect. Having a shorter time to first cigarette (i.e., smoking first cigarette within 5 minutes of waking) indicates greater nicotine dependence and thus, a greater HSI score. If HSI was calculated using the scoring in Table S1 it is incorrect and will need to be recalculated and any results related to HIS would need to be reinterpreted. Please correct the HSI scoring in Table S1 and the analyses/results if necessary.
RESPONSE:
WE THANK THE REVIEWER FOR HIGHLIGHTING THE ERROR. THE ERROR, HOWEVER, WAS ONLY IN THE SUPPLEMENTARY FILE WHILE THE CODING OF THE DATA WAS CORRECT. THUS, THE HSI SCORES AND CATEGORIES DID NOT CHANGE.
- Add a title/HSI score to #11
RESPONSE:
THE TITLE HAS BEEN ADDED.
- Clarify the meaning of the highlights or remove
RESPONSE:
THE HIGHLIGHTS HAS NOW BEEN REMOVED.
